# Parental Programming of Offspring Health: The Intricate Interplay between Diet, Environment, Reproduction and Development

**DOI:** 10.3390/biom12091289

**Published:** 2022-09-13

**Authors:** Vipul Batra, Emily Norman, Hannah L. Morgan, Adam J. Watkins

**Affiliations:** Lifespan and Population Health, School of Medicine, University of Nottingham, Nottingham NG7 2UH, UK

**Keywords:** parental environment, lifestyle factors, developmental programming, periconception period, animal models

## Abstract

As adults, our health can be influenced by a range of lifestyle and environmental factors, increasing the risk for developing a series of non-communicable diseases such as type 2 diabetes, heart disease and obesity. Over the past few decades, our understanding of how our adult health can be shaped by events occurring before birth has developed into a well-supported concept, the Developmental Origins of Health and Disease (DOHaD). Supported by epidemiological data and experimental studies, specific mechanisms have been defined linking environmental perturbations, disrupted fetal and neonatal development and adult ill-health. Originally, such studies focused on the significance of poor maternal health during pregnancy. However, the role of the father in directing the development and well-being of his offspring has come into recent focus. Whereas these studies identify the individual role of each parent in shaping the long-term health of their offspring, few studies have explored the combined influences of both parents on offspring well-being. Such understanding is necessary as parental influences on offspring development extend beyond the direct genetic contributions from the sperm and oocyte. This article reviews our current understanding of the parental contribution to offspring health, exploring some of the mechanisms linking parental well-being with gamete quality, embryo development and offspring health.

## 1. Introduction

It is widely accepted that our health and well-being as an adult are influenced by our immediate environment and lifestyle choices. However, it is becoming increasingly clear that our risk for developing a range of non-communicable diseases, such as obesity, type 2 diabetes and cardiovascular disease, are also shaped by the environment in which we developed prior to birth. There is now a significant body of experimental and epidemiological data linking maternal well-being during pregnancy with an increased risk of metabolic, cardiovascular, endocrine and behavioral ill-health in her adult offspring [1]. This association between development in utero and health and well-being in adult life defines the central concept of the Developmental Origins of Health and Disease (DOHaD) hypothesis [2]. During prenatal development, a range of factors, such as maternal nutrition, alcohol consumption, drug use and stress, can all interact with the genome of the developing embryo/fetus/offspring, resulting in multiple, distinct morphologic and physiologic adaptations [3]. It is hypothesized that these adaptations are made in an attempt to confer the offspring with a selective advantage in the anticipated postnatal environment [4]. Studies have shown that a range of fetal and neonatal adaptations are made in response to such factors, resulting in physiological manifestations such as reduced muscle fiber and cardiomyocyte proliferation [5], reduced pancreatic β-cell proliferation [6] and reduced kidney nephron deposition [7]. Whereas these in utero adaptations may confer on the offspring the opportunity of reaching sexual maturity and passing on their own genes to the next generation, their long-term consequences increase an individual’s predisposition toward adult ill health. However, it cannot be discounted that the observed changes in offspring phenotype are a reflection of the changes induced within the parental gametes and their direct impact on subsequent post-fertilization development. 

One interesting observation that emerged from the many early DOHaD studies was the realization that different periods of development displayed differential sensitivity to the impacts of a perturbed environment. Of critical importance was the observation that the periconception period (typically referred to as the period encompassing the final stages of gamete maturation, fertilization and preimplantation embryo development) appeared particularly vulnerable to suboptimal environmental conditions [8]. Such conditions can include variations in maternal nutrition, elevated or decreased gestational BMI, exposure to environmental pollutants, climate extremes, increased stress and exposure of gametes and embryos to suboptimal culture conditions *in vitro* [9]. Such factors can all impact the female’s fertility, resulting in a reduction in oocyte maturation, ovulation and quality [10,11]; reduced rates of fertilization and preimplantation embryo development; decreased numbers of cells within the embryo at the time of implantation [12]; perturbed rates of fetal development [13]; and an increased propensity for developing cardio-metabolic ill health in adulthood [1]. However, while there has been a significant focus on improving maternal health and well-being, the same has not been true for that of the father. The significance of his health and well-being at the time of conception is now a major focus of research in DOHaD [14]. Here, similar to that of the mother, studies have identified a connection between paternal health at the time of conception and sperm quality, embryo development and offspring health [15]. 

Interestingly, while it is becoming increasingly clear that either poor maternal or paternal health can negatively impact the long-term wellbeing of their offspring, few studies have analyzed the combined effect of both parents. Most studies have typically explored the impact of manipulating one parent in isolation from the other. However, this is in stark contrast to the situation for most people and animals, where the environmental and/or lifestyle conditions of one parent will be largely the same as for the other. As such, the use of a singular parental insult model lacks real-world relevance as couples are most likely to share lifestyle habits and diet choices [16]. In this review, we focus on the significance of periconception lifestyle factors in both parents with respect to the health of their offspring. The significance of the periconception period centers on the fact that it represents a time during which the gametes mature, their genomes undergo dramatic reorganization prior to global epigenetic remodeling and the first two distinct lineages of the embryo are established [17]. These lineages, the trophectoderm and the inner cell mass, ultimately form the placenta and fetus, respectively. Therefore, factors affecting these sensitive reproductive and developmental processes have the ability to influence offspring growth, development and future health. We explore both animal model and human data to understand the interaction between the parental environmental and lifestyle factors, gamete quality at the time of conception and the way they shape subsequent embryo development, fetal growth and offspring health. Understanding how poor parental lifestyle may impact on offspring development is of critical importance for improving global lifelong health.

## 2. Parental Diet and Obesity

The essential components of any conception and successful pregnancy start with the oocyte and sperm. These crucial cells undergo specific and coordinated development and maturation processes to produce fully differentiated and highly specialized cells. Like most somatic cells and tissues in the body, the gametes are also susceptible to the influences of poor lifestyle and environmental factors. One of the most widely studied parental lifestyle factors with respect to developmental programming is the influence of obesity. Numerous studies have highlighted the negative impact that obesity has on fundamental fertility in either parent [18,19]. In men, both obesity and elevated BMI are associated with decreased testosterone levels, paralleled by elevated levels of estrogen, leptin and insulin, resulting in a state of hypogonadism [20,21]. With regard to sperm quality, obesity is linked with membrane damage and compromised DNA integrity, which are connected to high levels of reactive oxygen species (ROS) from mitochondria with reduced activity [22,23]. Consequently, sperm from obese males display elevated levels of DNA fragmentation and abnormal chromatin condensation [24,25]. Furthermore, semen analyses have highlighted changes in the proteomic and metabolomic profiles of seminal fluid from obese men, which might contribute to the reported low semen volume, sperm concentration, motility and highly abnormal morphology [26,27]. 

Similar to the sperm, oocytes from obese females also appeared compromised in quality with reduced developmental potential. Oocytes collected from obese mice displayed high rates of meiotic spindle abnormalities, defective mitochondria with impaired membrane potential and increased levels of reactive oxygen species [28,29]. Furthermore, maternal obesity in mice resulted in elevated levels of lipid accumulation and distribution in oocytes [29,30], a factor associated with compromised quality. Further to the direct influences on the oocyte, changes in maternal nutrition can also affect the composition of the follicular fluid, predisposing the maturing oocyte to additional nutritional stress [8]. The pre-ovulatory follicular fluid from obese women has been shown to have increased levels of glucose, lactate and insulin [10]. Such changes in metabolic status have been linked to a reduced fertility [31] and poor embryo outcome [32]. Furthermore, elevated levels of leptin associated with obesity can inhibit LH-stimulated estradiol production in granulosa cells, as well as inhibit the insulin-induced ovarian steroidogenesis, thus impeding follicle development [33,34]. Mitochondrial function in oocytes from obese females also appears compromised, similar to the changes seen in sperm. This is critical as the mitochondria within the embryo are derived solely from the oocyte. Appropriately functioning mitochondria in the preimplantation embryo are crucial for successful fetal development. Minor reductions in mitochondrial activity in the early two-cell embryo resulted in significant changes in the expression of key genes involved in neuronal and brain function in fetal mice [35]. Separately, mouse models of maternal diabetes have shown that the oocyte is subject to increased cellular stress, resulting in abnormal mitochondrial morphology and failures in fetal development [36]. Maternal obesity is also associated with an increased risk of developing gestational diabetes. Here, in addition to the potential impacts of obesity itself, the mother also experiences placental dysfunction, insulin resistance and elevated levels of circulating metabolites, such as glucose [37]. The consequences of maternal gestational diabetes are evidenced as an increase in offspring birth weight (macrosomia), increased infant mortality and long-term risks of developing diabetes, obesity and cardiovascular disease for the offspring [38]. Due to the significant long-term impacts of gestational diabetes on both the mother’s and offspring’s heath, many studies have started to devise predictive methodologies with which to identify those females most at risk [39]. Such studies have focused on identifying specific biomarkers, such as changes in DNA methylation of specific genes (e.g., *SLC16A11*) [40], tissue proteome and transcriptome profiling [41]; elucidating and characterizing the biochemical cargo of extracellular vesicles [42]; and establishing optimal dietary regimens [43]. Interestingly, studies have indicated that multiple micronutrient supplementation in women from high-income countries is associated with an increased risk of developing gestational diabetes and fetal overgrowth [44,45]. One such micronutrient, iron, is widely taken by women during pregnancy to support the increased demand on the maternal cardiovascular system [46,47]. However, recent studies have identified that higher levels of circulatory and/or stored iron were linked to the development of gestational diabetes [48,49] and preeclampsia [50,51]—gestational conditions know to detrimentally impact fetal development and offspring health. Gestational iron supplementation has also been linked to increased weight at birth [52]. As such, the association between maternal diet, metabolic disorders, perturbed fetal growth and adult ill health continues to be a central focus of many studies of developmental programming. 

The negative impacts of maternal or paternal obesity on gamete quality are of significant importance. However, evidence also highlights the detrimental effects of combining obesity-compromised sperm with obesity-compromised oocytes on post-fertilization development and offspring health. In humans, when both parents were classified as obese, a delay in the time taken to reach the blastocyst stage was observed in embryos, which was greater than if only one of the parents was obese [53]. In addition, maternal, paternal and combined parental obesity have been reported to alter the trophectoderm and/or inner cell mass cell number ratios in the early embryo in mice [53,54,55]. Interestingly, studies have indicated that the negative impacts of parental obesity on post-fertilization development occur as a combination of the outcomes observed from when either parent alone was obese, rather than an amplification effect [56]. 

The postnatal health of the offspring has also been well studied with respect to parental nutrition. An increased risk of birth defects, such as orofacial clefts (loss of eyes and abnormal ears) and neural tube defects, has been reported in offspring when both the parents were overweight [57]. Furthermore, parental pre-conception obesity is predictive of an increased offspring BMI from childhood to adolescence and adolescence to adulthood [58]. This elevated BMI is associated with increased adiposity, dyslipidemia and decreased insulin sensitivity, which can be transmitted to subsequent generations [59]. Combined pre-conception parental obesity in humans has been found to result in elevated weight gain in the offspring from birth to 2 years of age, exceeding any weight gain observed in response to singular maternal or paternal obesity [60]. In addition to the weight gain impacts seen in response to parental obesity, studies have also identified elevated risks of metabolic disorders, such as non-alcoholic fatty acid liver disease. In mice, offspring from obese parents display an elevated expression of genes involved in lipogenesis (Sterol Regulatory Element Binding Protein (Srebp) and Fatty Acid Synthase (*Fas*)) and reduced expression of hepatic beta-oxidation genes (Peroxisome Proliferator-activated Receptor-alpha (*Ppar-α*) and carnitine palmitoyltransferase I (*Cpt-1*)) [61]. This study also reported that a combined parental high-fat diet significantly elevated plasma insulin levels and increased the expression of the central hepatic glucose metabolism genes (e.g., Phosphoenolpyruvate Carboxyl Kinase (*Pepck*) and Glucose-6-phosphatase (*G6pase*)). As in other studies, when both parents were obese, the extent of the programmed metabolic phenotypes were exacerbated, suggesting an additive parental impact [61]. In humans, birthweight has been correlated positively with both maternal and paternal BMI [62]. However, maternal BMI was found to have a greater influence than paternal BMI. These contrasting observations highlight the need for corroborating the effects observed in experimental models with those from human studies, as well as for further investigations involving combined paternal insults. 

## 3. Parental Age

In many countries, the average age at which individuals become parents has been steadily increasing [63]. Similar to other physiological systems, advancing parental age negatively impacts male and female reproduction. In mice, older females display reduced numbers of immunomodulatory cells (T-cell subsets) in the uterus during gestation [64], associated with reduced late gestation fetal growth and dysregulated immune cell profiles in the adult offspring. Separate studies in mice have identified that offspring born to older mothers had an increased incidence of neurological disorder, such as increased anxiety, compared to offspring from younger dams [65]. With regard to advancing paternal age, the results appear mixed on the precise impact it has on offspring well-being. Advanced paternal age has shown programming effects similar to those observed in advanced maternal age studies [66], with lower birthweight and associated health risks, as well as increased neurological problems, such as increased likelihood of developing autism [67,68]. The age of both parents is an important consideration as advanced maternal and paternal age has been associated with a longer time taken to pregnancy, suggesting that an advanced parental age reduces couple fecundity [69]. Interestingly, it is not just the absolute parental age that appears to have an impact on childhood development. A large epidemiological study found that a greater difference between the parents’ ages was associated with an increased likelihood of autism development in the offspring [70]. This observation highlights the possibility that a miss-matched maternal and paternal environment may be more detrimental to fetal development than a situation where both parents are equally impacted. 

## 4. Parental Exposure to Environmental Pollutants

A growing body of evidence suggests that periconception exposure to certain environmental pollutants and toxins detrimentally affects gamete quality, conception potential and embryonic development. As many of these chemicals are ubiquitous in our daily lives, as well as very stable in our environment, our exposure to them is increasing year after year. One major class of such environmental toxins are referred to as xenobiotics or xenohormones. These chemicals are of natural or synthetic origin and replicate our endogenous hormones, interfering and/or mimicking their actions. Hence, these chemicals have been termed Endocrine Disrupting Chemicals (EDCs) [71]. Studies have shown that exposure to such pollutants has a negative impact on male and female reproduction as well as post-fertilization development. In male rats, exposure to bisphenol-A (BPA) disrupts the hypothalamic-pituitary-gonadal axis, impairing testicular steroidogenesis and spermatogenesis resulting in sperm which undergo the acrosome reaction prematurely and which display low mitochondrial membrane potential [72,73]. Other studies have indicated that high BPA exposure induces a significant reduction in sperm concentration, vitality, count, motility and morphology and an increase in DNA damage in both mice and men [74,75]. Following conception, embryos derived from phthalate-exposed male mice have decreased implantation potential and lower chances of survival [76,77,78]. Paternal exposure to EDCs has also been shown to impact on postnatal offspring development. Children born to men exposed to pesticides containing phthalates are at a higher risk for cryptorchidism and hypospadias [79], while separate studies have observed a higher risk of being born small for gestational age in children of men exposed to phthalate and BPA [80,81].

Similar to males, female reproductive fitness is also negatively impacted by exposure to environmental pollutants. In women, elevated levels of urinary BPA correlated with decreased ovarian reserve and a lower likelihood of becoming pregnant [82]. In cows, oocytes from BPA-exposed females displayed a reduced ability to undergo maturation and had a higher degree of spindle malformations [83]. These abnormalities in bovine oocytes subsequently translated in to post-fertilization impairments, including reduced rates of post-fertilization embryonic development with increased rates of apoptosis [84]. Several studies have also indicated decreased rates of embryo implantation [85,86,87,88], abnormal embryo development [89,90] and differential expression of placental imprinted genes related to fetal growth and nutrition [91] in females exposed to pesticides, heavy metals and xenobiotics. Maternal exposure to chemicals, such as systemic fungicides and Vinclozolin during pregnancy, has also been shown to impact offspring fertility, inducing increased incidences of spermatogenic cell defects, testicular and prostate abnormalities as well as characteristics of polycystic ovarian disease in offspring [92]. Interestingly, studies have identified negative correlations between maternal and paternal phthalate exposure and placental and birth weights in their children [93]. Separately, the same group also found that maternal preconception urinary BPA concentrations, as well as paternal preconception urinary paraben concentrations, were prospectively associated with a higher risk of preterm birth [94]. However, only maternal BPA exposure was negatively associated with birth weight, with no association observed for the father [95]. 

## 5. Mechanisms Underlying Parental Programming

### 5.1. Parental Epigenetic Status

As the sperm and oocyte mature, they undergo dramatic epigenetic remodeling. The main consequence of these is the induction of a state of cellular quiescence with minimal transcription and, in the case of the sperm, a highly condense chromatin state [96]. Advances in high throughput analytical methods have allowed for the genome-wide mapping of these epigenetic changes during gamete maturation. As such, these techniques have been applied to understanding how environmental insults impact on these crucial epigenetic processes, identifying fundamental mechanisms for the germline intergenerational transmission of parental programming [97]. Maternal obesity in mice has been associated with a reduction in Stella protein abundance in oocytes, which results in global hypomethylation of the zygote and may be linked to impairments in fetal growth [98]. In humans, metastable epialleles in the oocyte appear to be affected by seasonal maternal nutritional status and such alterations in the epigenetic profile are still evident in childhood and adolescence, inducing persistent phenotypic changes in immune and metabolic function [99]. In sperm, similar epigenetic responses to poor preconception nutrition are observed as in the female. Sperm have a host of epigenetic-mediated mechanisms that can be altered by paternal health. These epigenetic marks are relatively well studied compared to those of the oocyte and have been found to have play key roles in regulating sperm function, as well as influencing development of the early embryo [100]. Of these epigenetic modifications, sperm DNA methylation status and its impacts on development are among the most detailed. Sperm DNA is known to be heavily methylated in normal male physiology and, in multiple human and animal models, the DNA methylation status can be disrupted by nutrition. In mice, a low-protein diet led to a global sperm DNA hypomethylation, which was not associated with a reduced fertility but with dysregulation of key cardiometabolic genes in the offspring [101,102]. This abnormal loss of DNA methylation has been associated with impaired embryo cell cleavage, abnormal blastocyst development and alterations in placental gene expression, ultimately impacting offspring cardiovascular and metabolic profiles in mice [102]. Advanced paternal age has also been reported to induce hypomethylation of sperm DNA [103], which has been linked to altered offspring behavior and neurological development in mice [104]. Furthermore, exposure to high-sugar diets in *Drosophila* has been shown to induce intergenerational metabolic reprogramming in offspring through conserved sperm chromatin-dependent signatures which were maintained in the offspring [105]. Sperm can also elicit epigenetic effects on embryo development due to their dynamic populations of small RNAs that are delivered to the oocyte upon fertilization [106]. Modifications to the sperm ‘RNA code’ in response to paternal environmental insults have been shown to influence the phenotype of the offspring [15]. In a mouse model of paternal high-fat diet (HFD), the total RNA content of sperm was found to be responsible for specific metabolic alterations in offspring phenotype, such as impaired glucose tolerance, stemming from early dysregulated gene expression in the blastocyst [107]. A similar mechanism involving changes to RNA profiles has been proposed in humans, as acute dietary changes such as a 1-week increase in sugar were sufficient to alter the patterns of sperm RNAs [108]. 

The dynamic nature of the germline epigenome may also allow for a reversal of any aberrations [109]. Paternal exercise in HFD-fed mice can return certain X-linked sperm miRNA expressions to control levels, which is associated with improvements to the metabolic phenotype of female offspring, which can be perturbed by paternal HFD alone [56]. Other studies, in both rodents and humans, have also shown that changes to sperm DNA methylation [110,111] and RNA composition, including miRNAs and tRNAs [112,113,114] can reduce the likelihood of poor offspring outcomes in HFD-fed or obese males. Likewise, maternal exercise interventions in mice have been found to prevent the HFD-induced hypermethylation of PGC-1 promoter in offspring skeletal muscles protecting the offspring from metabolic dysfunction observed with no exercise intervention [115]. However, it remains unclear whether exercise in one parent is able to negate the negative influences of poor nutrition in the other. 

### 5.2. Embryo Directed Mechanisms of Parental Programming

Epigenetic modifications in the early embryo play a major role in directing the programming of fetal and offspring development in response to parental stressors. Periconceptional environmental exposures can influence embryonic and placental expression of ncRNAs, genomic imprinting, DNA methylation and histone modifications [116,117,118]. In humans, the season of conception has been shown to alter the methylation status of the embryo in early development, with reduced methylation of identified differentially methylated regions (DMRs) in blastocysts associated with nutritional changes of the parents [99]. Similar epigenetic mechanisms underlie the detrimental effects of advanced paternal age or advanced maternal age on offspring development and health [119]. In male mice, an advancing age is associated with hypermethylation of the *Kcnq1ot1* imprinting control region. Hypermethylation leads to a dysregulation in the control of imprinted genes in the developing placenta, which is implicated in impaired growth, development and metabolism outcomes of the fetus [120]. Such alterations in the expression dynamics of placental genes, perturbation of epigenetic profiles and impaired growth suggest disconcerted developmental programming, which can negatively influence pregnancy outcome and offspring health [1,121].

Evidence from the use of assisted reproduction technologies (ART), whereby the embryo is exposed to an artificial environment *in vitro*, further highlights the sensitivity of the pre-implantation embryo to external environmental disturbances. ART-conceived children were found to have a smaller birthweight than normally conceived children, which was independent of various parental factors, suggesting that the exposures pre-implantation established slow in utero growth [122]. This may be due to altered gene expression in the early embryo due to suboptimal factors in the culture medium, which can influence cleavage rates and metabolism, which, in turn, can, impact fetal and placental development [123]. The ART-induced stresses on gametes and embryos have also been implicated with perturbations of developmental programming and an impairment of offspring health. However, the interpretation of the effects of these technologies on offspring outcome is complex since several demographic confounders exist, such as social, cultural, behavioral and educational status [124,125,126,127]. ART-induced changes to the epigenetic landscape of the developing embryo can be varied and tissue/gene-specific, which are then subject to masking by the post-natal environment [12,128,129]. 

### 5.3. Parental Influences on the Uterine Environment

The maternal uterine environment mediates early preimplantation embryo development as well as modulating implantation itself. Post-implantation variations in maternal exposures can also directly influence fetal in utero development. A perturbed metabolic, steroidal or immunological milieu during the periconceptional period may be one of the common mechanistic pathways via which distinct insults culminate in the development of poor health outcomes for the adult offspring [13]. As the developing embryo progresses through the female reproductive tract, it is exposed to a host of bioactive factors, such as chemokines, cytokines and growth factors, which can themselves be altered in response to environmental stressors, e.g., age and dietary habits. For example, maternal obesity is associated with an increase in circulating leptin [130]; therefore, the developing embryo is exposed to these elevated levels. This can result in a hypothalamic response to leptin and subsequent modulation of hunger and pancreatic β-cell physiology in the fetus [131]. 

Changes in uterine luminal components can be dictated by paternal seminal fluid, which can also influence developmental programming effects in the progeny. The insemination or introduction of semen derived factors into the uterus triggers a myriad of immune responses which assist the spermatozoa in fertilization and aid implantation [132,133]. Endometrial inflammation is driven by seminal factors, such as prostaglandins, TGFβ and interleukins, which can stimulate the production of endometrial chemokines/cytokines and recruit immune cells that activate inflammatory pathways necessary for decidual tissue remodeling [133,134,135]. Further maternal immune responses are directed by paternal antigens in the seminal plasma, which are essential for immune tolerance of the developing embryo [136,137]. The seminal plasma has increasingly been accepted as a non-genetic medium for the transmission of paternal influences to the offspring [138]. The composition of seminal plasma not only directs uterine immune responses but has also been found to impact placental, embryonic and offspring development [139,140]. Factors within the seminal fluid are key to embryo development, as total removal of the plasma component in mice results in severely impaired preimplantation embryo development [138]. A paternal high-fat diet has been shown to disrupt the uterine milieu by dysregulation the production of TGF-β, IL-10 and TNF in the seminal fluid, which was then found to disrupt the uterine inflammatory gene expression profiles [141]. The full profile of embryo-trophic factors in the seminal plasma has not yet been confirmed; however, an imbalance in uterine/oviductal signaling molecules during preimplantation development can affect the embryo’s metabolic status, which could have implications for future fetal development and offspring outcome [142,143,144]. In the complete absence of seminal fluid, embryo development rate is significantly reduced. Moreover, of the reduced number that produce a viable fetus, these offspring are significantly heavier than their counterparts exposed to seminal fluid in those early developmental stages [138]. Furthermore, paternal diet may also alter seminal fluids’ roles in uterine mediation and embryo development. A study in mice found that a control embryo (generated using sperm from control-diet fed males) exposed to seminal fluid from a low-protein-diet-fed male resulted in a lasting metabolic programming effect in the offspring, demonstrated by an increased post-natal weight as well as disruptions to glucose tolerance and hepatic glucose regulation genes [102]. The interplay between seminal fluid, embryo and uterine receptivity highlights the importance of seminal fluid composition in implantation. This entails extensive, intricate and highly coordinated biochemical, physiological and morphological cellular processes, which, if perturbed by a poor paternal environment, can be detrimental to fetal development and offspring phenotype [145,146,147]. However, while some bioactive seminal fluid factors have been identified as key regulators of embryo development, the full profile of factors is still elusive. Identifying these factors is important in obtaining a full picture of how seminal fluid is influencing the maternal environment and how these interactions may be altered by parental environmental challenges.

## 6. Conclusions and Future Perspectives

Over the past 40 years, a new realization that our adult health can be shaped and influenced by how we developed in the womb prior to birth has emerged. Furthermore, we have come to understand that factors prior to conception, such as the quality of the parental gametes, can impact the normal developmental processes regulating the earliest stages of life (Figure 1). These studies have provided evidence of the connection between parental well-being at the time of conception and the transmission of acquired traits to their offspring (see Table 1). Such observations provide an understanding into the recent global rise in the prevalence of non-communicable diseases, identifying a range of mechanisms that can link gamete quality with trans-generational ill health. Furthermore, these parental influences on offspring well-being appear maintained, even after the parents resume a normal diet [148]. Such observations have significant implications for human reproduction and health, where parents will typically improve their lifestyle habits in the months immediately prior to conception. However, while these connections have been identified, many knowledge gaps still remain. Here, the impact of other parental factors, such as stress, have yet to be explored fully. Furthermore, as highlighted in this review, relatively few studies have yet to define the programming influences of both parents together. As such, more investigation into the interaction between maternal and paternal contributions to childhood outcomes are needed, shifting the focus to both parents rather than either one in isolation. 

## Figures and Tables

**Figure 1 biomolecules-12-01289-f001:**
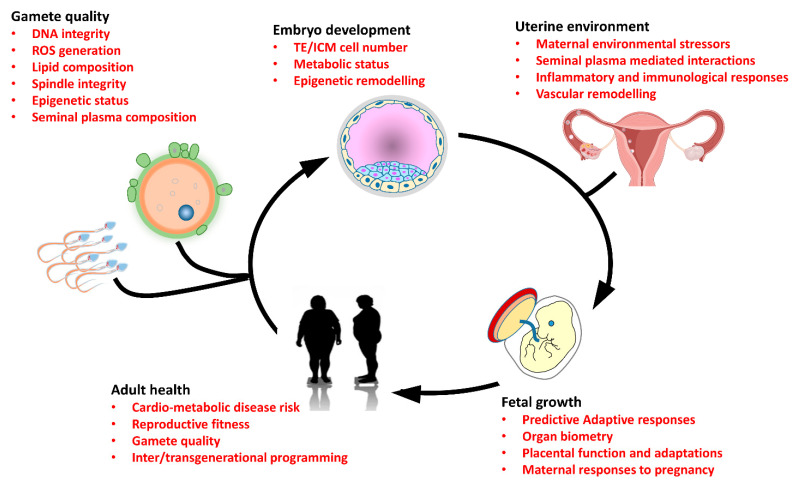
Diagram depicting the central factors linking parental environmental and lifestyle aspects with gamete quality, preimplantation embryo development, fetal growth and adult offspring health. Exposure of parents to poor-quality diet, environmental toxins and pollutants or advancing age can all negatively impact gamete quality, impairing DNA integrity and elevating levels of damaging reactive oxygen species (ROS). Poor-quality gametes at the time of conception subsequently perturb preimplantation embryo development, altering the genome-wide epigenetic remodeling that takes place and affecting cell numbers within the initial embryo lineages, the inner cell mass (ICM) and trophectoderm (TE). Within the uterine environment, factors such as maternal diet, pollution, advancing age and/or stress can directly impact the developmental dynamics and long-term trajectory of the preimplantation embryo through alterations in immune cell populations. In addition, paternal seminal plasma components have also been shown to influence the uterine environment, modulating the proportions of immune cells, as well as inflammatory responses and vascular remodeling, all essential for successful embryo implantation. In late gestation, differential fetal growth can occur through perturbed nutrient provision and dysfunctional placental transport. Here, disproportional fetal growth dynamics result in significant changes in tissue structure and function. These fetal predictive adaptive responses can then predispose the adult offspring to an increased risk of developing a range of non-communicable cardio-metabolic diseases in adult life, which can impact on the quality of their own gametes and the health of subsequent generations.

**Table 1 biomolecules-12-01289-t001:** Summary of studies that found altered fetal/offspring outcomes in response to singular (maternal or paternal) or combined parental exposures.

Factor	Singular/Combined	Major Outcomes/Findings	Species	References
Parental obesity	Paternal alone	Minimal impacts on seminal quality. Some studies indicate links between the Kisspeptin signaling pathway and obesity-induced male infertility Obesity may disturb early embryonic cell cycles kinetics, seminal/spermatozoal biomolecular composition, diminish reproductive performance and increase sperm oxidative stress	Human/Rodents	[20,21,22,23,24,25,26,27]
	Maternal alone	Maternal obesity been associated with high rates of meiotic spindle abnormalities, defective mitochondria, increased levels of reactive oxygen species, dyslipidemia, perturbed follicular fluid composition and oocyte quality	Humans/Mice	[28,29,30,31,32,33,34,35,36]
	Combined parental	Combined parental obesity leads to impaired post-fertilization development and offspring health, e.g., increased risk of birth defects, elevated weight gain in postnatal life and metabolic disorders in adulthood	Humans/Mice	[53,54,55,56,57,58,59,60,61,62]
Advanced parental age	Paternal alone	Perturbed growth and development, dysregulated immune cell profiles, increased risk of autism spectrum disorders and impaired neurocognition	Humans/Mice	[66,67,68]
	Maternal alone	Programming effects similar to those observed in advanced paternal age studies	Mice	[64,65]
	Combined parental	Reduced fecundity and compromised neuro-development	Humans	[69,70]
Parental Exposure to Environmental Pollutants	Paternal alone	EDCs disrupt the hypothalamic-pituitary-gonadal axis, thus compromising sperm/semen quality, conception potential and embryonic development	Humans/Rodents	[72,73,74,75,76,77,79]
	Maternal alone	EDCs disrupt the hypothalamic-pituitary-gonadal axis, affecting oocyte’s structural and functional integrity and thus compromising fertilization potential	Humans/Mice/Cattle	[82,83,84,85,86,87,88,89,90,91,92]
	Combined parental	Adverse pregnancy outcomes, e.g., altered morphometry of fetus/embryo, impaired growth	Humans	[78,80,81,93,94,95]

EDCs: endocrine-disrupting chemicals.

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
