# Peer review of "Parental Programming of Offspring Health: The Intricate Interplay between Diet, Environment, Reproduction and Development"

_biomolecules, 2022, doi:10.3390/biom12091289_

Round 1

Reviewer 1 Report

In this manuscript, Batra et al review the interplay between diet, environment, reproduction and development and the impact that parental health can have on subsequent generations. Overall, this is a beautifully written review that covers a complicated topic in detail. Importantly, this team is at the forefront of this area of research and are well suited to cover this important topic. I should note that the references were not attached to the article so I was unable to determine whether there was self-citation or inappropriate referencing used. This should be checked by an editor/reviewer prior to acceptance. In my mind, this article is perfectly suited for publication in this journal. I do have a few minor comments for consideration:

1. Line 40-41 - "These adaptions are made in an attempt to confer the offspring with a selective advantage in the anticipated post-natal environment" - While I appreciate this may very well be true in many cases, if the environmental exposure causes damage to gametes, will the altered phenotype also be a reflection of conferring a selective advantage?? Do we know that all adaptions to environmental exposures confer a selective advantage or could it be a marker of damage??

2. Gene title nomenclature: This is largely exactly as it should be but I did notice in line 157, Pepck is not italicised. Authors should carefully check to make sure all genes follow appropriate nomenclature guidelines

3. Line 168-170: "In mice, older females display reduced numbers of immunomodulatory cells in the uterus during gestation, resulting in poor fetal growth and dysregulated immune cell profiles in offspring" I apologise for this one but as I wasn't able to check the reference, I could not confirm. Did the authors in this manuscript show that the altered immune cells in the uterus were the specific cause of dysregulated immune cell profiles in offspring?? If not, the authors should carefully change the wording to highlight that you speculate that these may be linked but further studies are required to specifically determine this.

4. The authors highlight the importance of studying the combined influence of both mothers and fathers on the inheritance of traits in offspring. This is a nice addition to this manuscript and highlights an important area that needs to be further studied. The authors do indeed touch on this throughout the manuscript but perhaps this could be better supported by a summary table for each exposure which shows what are the altered outcomes for maternal, paternal and combined exposures, and the species these studies are completed in. This would highlight the areas where further research may be required and would be a nice addition.

5. Figure 1 - This is a nice summary figure. The authors should consider adding in the uterine environment as a separate component. Currently seminal plasma composition comes under gamete quality, but it also affects this environment. Additionally maternal exposures have potential to affect the uterine environment as well so this would be a valuable addition. 

Author Response

Biomolecules-1871648

 Reviewer #1

In this manuscript, Batra et al review the interplay between diet, environment, reproduction and development and the impact that parental health can have on subsequent generations. Overall, this is a beautifully written review that covers a complicated topic in detail. Importantly, this team is at the forefront of this area of research and are well suited to cover this important topic. I should note that the references were not attached to the article so I was unable to determine whether there was self-citation or inappropriate referencing used. This should be checked by an editor/reviewer prior to acceptance. In my mind, this article is perfectly suited for publication in this journal. I do have a few minor comments for consideration:

Response: We thank the reviewer for their helpful comments and their positive assessment of our manuscript. We apologise that the References were not included in the version of the manuscript that you received. We have checked and they were included in the WORD version of the manuscript that we submitted to the journal. However, it seems that the reviewers were sent a heavily formatted version of the manuscript which did not have the references attached. We are unsure why this was the case and will raise it in our covering letter.

  1. Line 40-41 - "These adaptations are made in an attempt to confer the offspring with a selective advantage in the anticipated post-natal environment" - While I appreciate this may very well be true in many cases, if the environmental exposure causes damage to gametes, will the altered phenotype also be a reflection of conferring a selective advantage?? Do we know that all adaptions to environmental exposures confer a selective advantage or could it be a marker of damage??

Response: The Reviewer raises a very insightful and interesting comment. We have amended the text to indicate that the predictive adaptive responses are mainly hypothesised, please see lines 54-55. In addition, we have added in some additional text to state that the changes in offspring phenotype could be a direct reflection of any changes in gamete quality and status. Please see lines 61-63:-

            ‘However, it cannot be discounted that the changes in offspring phenotype observed are a reflection of the changes induced within the parental gametes and their direct impact on subsequent post-fertilisation development’. 

  1. Gene title nomenclature: This is largely exactly as it should be but I did notice in line 157, Pepck is not italicised. Authors should carefully check to make sure all genes follow appropriate nomenclature guidelines

Response: We thank the Reviewer for identifying the mistake with the gene styling for Pepck. This has now been corrected.

  1. Line 168-170: "In mice, older females display reduced numbers of immunomodulatory cells in the uterus during gestation, resulting in poor fetal growth and dysregulated immune cell profiles in offspring" I apologise for this one but as I wasn't able to check the reference, I could not confirm. Did the authors in this manuscript show that the altered immune cells in the uterus were the specific cause of dysregulated immune cell profiles in offspring?? If not, the authors should carefully change the wording to highlight that you speculate that these may be linked but further studies are required to specifically determine this.

Response: As above, we apologise to the Reviewer that they were unable to see which reference this sentence referred to. We hope that subsequent revisions of this manuscript will contain the reference sections. Just for the Reviewer’s own interest, the reference at this point was Levenson D et al., The effects of advanced maternal age on T-cell subsets at the maternal-fetal interface prior to term labor and in the offspring: a mouse study. Clinical and experimental immunology 2020, 201(1):58-75.

The reviewer is correct that the study only made associations between the changes in uterine immune cell populations and differential fetal growth and immune cell profiles. As such, we have amended the text as requested to make it clear that the uterine immune profiles were associated with changes in fetal growth and adult immune cell populations. Please see lines 194-196:-

            ‘In mice, older females display reduced numbers of immunomodulatory cells (T-cell subsets) in the uterus during gestation [64], associated with reduced late gestation fetal growth and dysregulated immune cell profiles in the adult offspring’

  1. The authors highlight the importance of studying the combined influence of both mothers and fathers on the inheritance of traits in offspring. This is a nice addition to this manuscript and highlights an important area that needs to be further studied. The authors do indeed touch on this throughout the manuscript but perhaps this could be better supported by a summary table for each exposure which shows what are the altered outcomes for maternal, paternal and combined exposures, and the species these studies are completed in. This would highlight the areas where further research may be required and would be a nice addition.

Response: A summary table depicting the individual and combined influence of both mothers and fathers exposures on the inheritance of traits in offspring has now been added, see Table 1.

  1. Figure 1 - This is a nice summary figure. The authors should consider adding in the uterine environment as a separate component. Currently seminal plasma composition comes under gamete quality, but it also affects this environment. Additionally maternal exposures have potential to affect the uterine environment as well so this would be a valuable addition. 

Response: We thank the Reviewer for their praise of the figure and for highlighting the maternal omission. We have amended the figure to include additional information on the uterine environment impacts and the role of maternal exposures. This detail has also been added to the figure legend

Reviewer 2 Report

This is an interesting and well written article on the impact of parental programming on offspring health. The contributions of both father and mother are addressed. However, there are a few important considerations: 

1. The references were not included in the version that was available for review, so it is hard to know which key references were included. 

2. The impact of paternal diet has been well studied however it is unclear if these studies were referenced:

 a) Paternal diet defines offspring chromatin state and intergenerational obesity Öst et al 2014, 

  b) Multigenerational Undernutrition Increases Susceptibility to Obesity and Diabetes that Is Not Reversed after Dietary Recuperation, Hardikar et al 2015

3. The impact of maternal exposures like GDM and anemia on fetal programming needs to be included (Mapping the Cord Blood Transcriptome of Pregnancies Affected by Early Maternal Anemia to Identify Signatures of Fetal Programming, Hatem et al 2022, Downregulation of SLC16A11 is Present in Offspring of Mothers with Gestational Diabetes Sevilla-Domingo 2022 and many others)

Author Response

Reviewer #2

This is an interesting and well written article on the impact of parental programming on offspring health. The contributions of both father and mother are addressed. However, there are a few important considerations:

Response: We thank the Reviewer for their consideration of our article and for their comments.

  1. The references were not included in the version that was available for review, so it is hard to know which key references were included.

Response: We apologise to the Reviewer that the References were not included in the version of the manuscript that you received. We have checked and they were included in the WORD version of the manuscript that we submitted to the journal. However, it seems that the reviewers were sent a heavily formatted version of the manuscript which did not have the references attached. We are unsure why this was the case and will raise it in our covering letter.

  1. The impact of paternal diet has been well studied however it is unclear if these studies were referenced:

  1. a) Paternal diet defines offspring chromatin state and intergenerational obesity Öst et al 2014,
  2. b) Multigenerational Undernutrition Increases Susceptibility to Obesity and Diabetes that Is Not Reversed after Dietary Recuperation, Hardikar et al 2015

Response: we thank the reviewer for their suggested additional references. These were not originally included in our references but have now been incorporated, please see lines 277-279 and lines 387-390.

            ‘Furthermore, exposure to high sugar diets in Drosophila has been shown to induce intergenerational metabolic reprogramming in offspring through conserved sperm chromatin-depend signatures which were maintained in the offspring [89]’

            ‘Furthermore, these parental influences on offspring well-being appear maintained, even after the parents resume a normal diet [132]. Such observations have significant implications for human reproduction and health were typically parents will improve their lifestyle habits, but only in the few months prior to conception.    

  1. The impact of maternal exposures like GDM and anemia on fetal programming needs to be included (Mapping the Cord Blood Transcriptome of Pregnancies Affected by Early Maternal Anemia to Identify Signatures of Fetal Programming, Hatem et al 2022, Downregulation of SLC16A11 is Present in Offspring of Mothers with Gestational Diabetes Sevilla-Domingo 2022 and many others)

Response: We thank the Reviewer for their suggestion with regard to additional topics such as gestational diabetes and maternal anaemia. While we agree with the Reviewer that their impact on fetal development are significant, the main focus of our Review was to cover topics that can affect both the mother and the father and, where possible, to discuss the impact of combined parental exposure. As such, there are no equivalent paternal gestational diabetes models (other than the impact of obesity) or models of gestational anaemia. Therefore, we would only be able to discuss the maternal impacts, which is not in line with the focus of our study. However, as we agree that these are significant factors, we have made reference to them within the body of the manuscript. Please see lines 136-156

Round 2

Reviewer 2 Report

All my concerns have been addressed. I have no further feedback.